# Bioconversion of Geniposide from *Gardenia jasminoides* via *Levilactobacillus* Enhancing Anti-Inflammatory Activity

**DOI:** 10.3390/foods14234156

**Published:** 2025-12-03

**Authors:** Chun-Zhi Jin, Long Jin, Ye Zhuo, Ting Li, Huimin Liu, Kee-Sun Shin, Le Kang

**Affiliations:** 1Key Laboratory of Nanchong City of Ecological Environment Protection and Pollution Prevention in Jialing River Basin, College of Environmental Science and Engineering, China West Normal University, Nanchong 637002, China; chunsik11111@gmail.com (C.-Z.J.); tingting93@cwnu.edu.cn (T.L.); liuhuimin@stu.cwnu.edu.cn (H.L.); 2Microbiome Convergence Research Center, Korea Research Institute of Bioscience and Biotechnology (KRIBB), 125 Gwahak-ro, Yuseong-gu, Daejeon 305-806, Republic of Korea; 3College of Ecology and Environment, Nanjing Forestry University, 159 Longpan Road, Nanjing 210037, China; isacckim@alumni.kaist.ac.kr; 4Cell Factory Research Center, Korea Research Institute of Bioscience and Biotechnology (KRIBB), 125 Gwahak-ro, Yuseong-gu, Daejeon 305-806, Republic of Korea; zhuoye123@kribb.re.kr

**Keywords:** *Levilactobacillus*, geniposide, bioconversion, *Gardenia jasminoides*, anti-inflammatory

## Abstract

Genipin, one of *Gardenia jasminoides’* bioactive components, exhibits superior therapeutic efficacy compared to geniposide, though it is present in much lower concentrations. Conventional hydrolysis methods using acids or organic solvents can enhance genipin yield but often raise environmental and safety concerns. This study aimed to increase genipin production through whole-cell bioconversion of geniposide to genipin using lactic acid bacteria (LABs). A total of 191 LAB strains were isolated from kimchi. *Levilactobacillus* sp. LN180102 showed the highest bioconversion activity, which was up to 40%. Docking analysis and esculin assay confirmed the beta-glucosidase activity. The anti-inflammatory effects of the fermented extract were enhanced by 28.5% in RAW 264.7 cells in vitro. Additionally, *Levilactobacillus* sp. LN180102 is probiotic-friendly and exhibits a high tolerance for phenol, bile, and acid. In their entirety, these discoveries have the potential to illuminate the ways in which *Gardenia jasminoides* can be functionally improved through whole-cell bioconversion, thereby enabling individuals to lead healthier lifestyles.

## 1. Introduction

Functional food has become a popular trend in modern life, providing not only nourishment as food but also specific health benefits for human beings [1,2]. *Gardenia jasminoides* Ellis (*Rubiaceae*) is a flowering plant (white flower), evergreen shrub of tropical or sub-tropical origins, mainly cultivated in China, Japan, Vietnam, and South Korea [3]. For thousands of years, it has been a well-known component of traditional herbal medicine in some Asian countries and is currently the focus of functional food research due to its diverse health benefits, including anti-obesity [4], anti-diabetic [5], antioxidant [6], sleep-improving [7], and anti-inflammatory [8] properties.

Inflammation is a primordial response that protects against infection and repairs damaged tissue to its normal physiological functioning [9]. Recent studies have emphasized the importance of maintaining inflammatory homeostasis and regulating the immune system. Inflammation is initiated by various immune cells, such as macrophages, to protect against harmful stimuli, such as viruses and bacteria [10]. During the inflammatory reaction, macrophages release mediators, like nitric oxide (NO), cytokines, and growth factors [11]. NO is essential for host innate immune responses to pathogens such as bacteria, viruses, and fungi, and plays an important role in controlling physiology, including neurotransmission, vasodilation, and neurotoxicity [12]. However, excessive production of NO results in the development of inflammatory diseases, like rheumatoid arthritis and autoimmune diseases. Therefore, suppression of NO production is considered to be important in the development of therapeutic agents for inflammatory diseases [13].

Geniposide is the main effective iridoid glycoside in *G. jasminoides* Ellis, and its content is approximately 3–8%, which varies depending on its origin [14]. Genipin, structurally the aglycon part of geniposide, could not receive much attention in functional research due to its low concentration, except for some known anti-inflammatory [15], cross-linker [16], and anti-cancer properties [17]. And generally, it is being reported that the function of genipin is much better than geniposide, like anti-inflammatory [15], anti-helicobacterium [18], *E. coli* [19], protection of Hepatic cell [20], and so on. Geniposide is typically consumed in its entirety from the fruit of *G. jasminoides*. Some individuals could naturally convert geniposide to genipin through intestinal microbial digestion, while others lack the necessary microbial activity. The key enzyme involved in this bioconversion is the beta-glucosidase from the microbe. For individuals without this capability, the use of probiotics, which are generally recognized as safe (GRAS) by regulatory organizations such as EFSA and FDA, can offer an optimal solution to increase genipin content [21]. Probiotic-mediated whole-cell bioconversion is a safe, environmentally friendly, and economically viable method for converting substrates from herbal medicinal plants into useful products, and nowadays it is widely used for evaluating functional foods and pharmaceuticals [22,23,24].

*Levilactobacillus* sp., a Gram-positive, rod-shaped, and hetero-fermentative potential probiotic microorganism, is found in a variety of fermented foods, including beverages, wine, beer, sourdoughs, meat, kimchi, and dairy products [25]. Lee’s group utilized *Levilactobacillus* casei KFRI 126, *Levilactobacillus curvatus* KFRI 166, and *Levilactobacillus confuses* KFRI 227 for bioconversion, achieving a 16% conversion ratio [26]. Kim’s group used an enzyme from *Levilactobacillus antri* to bio-convert it 100% [27], while Pan’s group fed *Levilactobacillus plantarum* KFY02 and geniposide to mice, resulting in in vivo conversion in the gut. And then there is no more report using probiotics for the bioconversion of geniposide to genipin. On the contrary, so many works are performed by using enzymes from fungi with a high bioconversion ratio [28,29,30,31,32]. Probiotic-mediated whole-cell bioconversion can modify the geniposide-to-genipin ratio, potentially enhancing the pharmaceutical functions of *G. jasminoides*, particularly its anti-inflammatory, anti-diabetic, anti-cancer, and other properties.

In this study, we developed a lactic acid bacteria library by isolating microorganisms from a Korean traditional food (*kimchi*) and identified a candidate with bioconversion potential for *G. jasminoides*. We subsequently performed studies to ascertain if the bioconversion of geniposide to genipin influences its anti-inflammatory effect in RAW 264.7 cells. Furthermore, we examined the probiotic potential of the chosen strain, *Levilactobacillus* sp. LN180102, which encompasses its overall physiological properties, resistance to gastric acidity, bile, phenols, antibiotics, and antimicrobial activities. It is the first report characterizing the probiotic characteristics of a LAB that could undergo bioconversion by *G. jasminoides*. Genome analysis validated the bioconversion by docking geniposide with beta-glucosidase. The application of *Levilactobacillus* sp. LN180102 for the bioconversion of *G. jasminoides* to augment its anti-inflammatory properties signifies a viable alternative for the advancement of functional foods.

## 2. Materials and Methods

### 2.1. Chemicals and Reagents

Dried fruits of *G. jasminoides* were obtained from Jecheon herb medicine mall, which was planted in Jindo-eup, Jindo-gun, Jeonnam, Republic of Korea. The herbal plants were grown in the Jecheon area. The dried herbal medicine was homogenized into powder form by homogenizer. Standard compounds, geniposide (≥95.0% purity) and genipin (≥97.0% purity) were purchased from Tokyo Chemical Industry Co., Ltd. (Tokyo, Japan). Geniposide and genipin were dissolved in water (HPLC grade) at 1 mg/mL. Then they were diluted and used to establish a standard curve. Ethanol (≥95% purity), methanol (≥99.9% purity), trifluoroacetic acid (≥99.0% purity), 3-(4,5-dimethyl-thiazol-2-yl)-2,5-diphenyl-tetrazolium bromide (MTT), and lipopolysaccharide (LPS) were obtained from Sigma-Aldrich, St. Louis, MO, USA. Man, Rogosa, Sharpe (MRS) medium was supplied by BD Difco, and fetal bovine serum (FBS), penicillin/streptomycin, and DMEM were purchased from Gibco BRL, Grand Island, NY, USA.

### 2.2. LABs Isolation and Screening for β-Glucosidase Activity

Fifty different homemade *kimchi* samples were collected from five cities (Seoul, Gangneung, Daejeon, Pohang, Jeju) in Republic of Korea. These samples were homogenized and serially diluted to a concentration of 10^−7^ in distilled water and incubated at 37 °C for 2 days after spreading on MRS agar media. Pure colonies were isolated and stored in glycerol (20%, *v*/*v*) at —80 °C for further analysis. Esculin—MRS media was composed of 1 g esculin, 0.5 g ferric ammonium citrate, and 55 g MRS in 1 L of water, pH 6.5. Finally, 0.1% LN180102 cells were inoculated and cultured in 5 mL Esculin—MRS media at 28 °C for 24 h, and then their color change was observed.

### 2.3. Bioconversion and Compounds Identification

Lactic acid bacteria were cultured in 10 mL liquid MRS medium at 37 °C for 24 h. Then cell cultures were centrifuged at 4000 rpm for 15 min to separate cells from the broth. The cell pellets were suspended in 1 mL saline solution, mixed with *G. jasminoides* water extract, and incubated at 37 °C for 48 h. To prepare the water extract, 1 g of *G. jasminoides* powder was mixed with 10 mL of water and autoclaved at 121 °C for 15 min. Bioconversion was assessed by monitoring peak changes from high-pressure liquid chromatography (HPLC) profiles of fermented and non-fermented *G. jasminoides*. The HPLC system (Hitachi, Tokyo, Japan) was equipped with a binary pump delivery system, a 1430 diode array detector, a 1210 autosampler, and a TSK-GEL C18 column (4.6 × 150 mm, 2.7 μm; Tosoh, Tokyo, Japan). The injection volume was 10 μL, and the mobile phase consisted of methanol (solvent A) and 0.04% trifluoroacetic acid in water (solvent B). The linear gradient elution program was as follows: 15% A for 0–5 min, 15–100% B for 5–20 min, 100% A for 20–25 min, 100–15% A for 25–35 min, and 15% A for 35–40 min. The flow rate was 1.0 mL/min, and the absorbance was detected at 240 nm [33]. Compounds were separated using an LH20 open column (Merck, Darmstadt, Germany) and eluted with 100% methanol. Fractions were purified using thin-layer chromatography (TLC). The isolated compounds were dissolved in HPLC-grade methanol (200 μg/mL) for LC/MS analysis. LC/MS was equipped with phenomenex 1260 infinity high-pressure liquid chromatography (Agilent Technologies, Santa Clara, CA, USA) and an AB sciex 3200 Q trap system (AB Sciex, Framingham, MA, USA). Samples were injected and separated by the HPLC system at a flow rate of 400 μL/min with 0.1% formic acid (FA) in acetonitrile (A) and 0.1% FA in water (B). The linear gradient elution program was as follows: 5–100% A for 0–7 min, 100% A for 7–10 min, 100–5% A for 10–20 min. Luna^®^ C18 (2) 100 Å (2.0 × 100 mm, 3 µm) column temperature was set at 25 °C. The scan range was m/z 100–1000, and the scan time was 20.0 min.

### 2.4. Taxonomic Analysis of Levilactobacillus sp. LN180102

Universal primers 27F (5′-AGA GTT TGA TCC TGG CTC AG-3′) and 1492R (5′-GGT TAC CTT GTT ACG ACT T-3′) were used for sequencing the 16S rRNA gene [34]. Sequences were compiled and edited using SeqMan software version 4.1 (DNASTAR Inc.) and MEGA version 7.0 [35]. A phylogenetic tree was constructed using the neighbor-joining algorithm, and the topology was assessed by bootstrap analyses of 1000 replications. Reference strains’ sequences were retrieved from the EzTaxon database (http://www.ezbiocloud.net/eztaxon, accessed on 18 March 2025).

### 2.5. Probiotic Characteristics of Lactic Acid Bacteria

The growth conditions for *Levilactobacillus* sp. LN180102 were tested, including temperature range (4.0–55.0 °C) and salt tolerance (0–10%, *w*/*v*). Antibiotic resistance was tested using an E-test strip containing ampicillin, vancomycin, gentamicin, kanamycin, streptomycin, clindamycin, erythromycin, tetracycline, chloramphenicol, or ciprofloxacin, following the European Food Safety Authority (EFSA) guidelines. The strain was cultured in MRS media for 24 h at 37 °C, and 100 μL of cells were spread onto MRS agar. The E-test strip was placed on agar media and incubated at 37 °C for 24–48 h. Hemolytic activity was assessed on a medium composed of tryptose (10.0 g), Lab-Lamco powder (3.0 g), sodium chloride (5.0 g), sheep blood (50 mL), agar (12.0 g), and the final pH was adjusted to 7.3. Cells were cultured on medium for 18–72 h at 35 °C, and the lysis of red blood cells was observed, following the method described by Talib et al. [36]. The isolate was subjected to MRS broth (pH 2.0) for 1-3 h. For bile tolerance, strain LN180102 was incubated in modified MRS containing 0.3% bile for 2, 4, 6, and 8 h. Phenol tolerance was assessed by inoculating the strain into MRS broth containing 0.1 and 0.4% phenol for 24 h. The cell suspensions were plated on MRS agar and incubated at 37 °C, 24 h for acid, bile, and phenol tolerance tests to determine colony-forming unit (CFU) [37]. The antimicrobial activity of *Levilactobacillus* sp. LN180102 was assessed using agar-well diffusion and agar-spot assays. The strain was cultivated in liquid MRS medium at 37 °C for 24 h. Then cells and broth were separated by centrifugation at 4000 rpm for 15 min. The pH of the supernatant was adjusted to 7.0 using 2N NaOH, and the adjusted supernatant was sterilized using a 0.2 µm pore-size filter. Cell suspension and broth of *Levilactobacillus* sp. LN180102 was either spotted on or loaded into BHI agar plates, which had been inoculated with pathogens, respectively. The plates were incubated until visible inhibition zones appeared, indicating antimicrobial activity. The pathogens tested in this study included *Staphylococcus aureus* KCTC 1621^T^, *Listeria monocytogenes* KCTC 3569^T^, *Streptococcus mutans* KCTC3065^T^, *Streptococcus salivarius* ATG-P1^T^, *Escherichia coli* KCTC 1682^T^, *Pseudomonas aeruginosa* KCTC 2004^T^, and *Cronobacter sakzakii* KCTC 2949^T^.

### 2.6. Cell Viability Assay and NO Assay

Mouse macrophage RAW 264.7 cells were supplied by the Kang lab, originally purchased from Hycyte company in Chengdu, China, and cultured in 10% FBS containing RPMI media supplemented with 1% penicillin-streptomycin-glutamine (Invitrogen) and maintained in a 5% CO_2_ incubator at 37 °C. Cell viability was assessed using the MTT assay. Cells were treated with ethanol extracts (50, 100 μg/mL), geniposide, genipin (20 and 40 μM), Dexa (10 μM), and L-NMAE (100 μM), which were dissolved in DMSO and incubated for 24 h. Absorbance was measured at 570 nm using a microplate reader. For the NO production assay, the concentration of Dexamethasone (Dexa) and N-nitro-L-arginine methyl ester (L-NAME) were up to 40 μM and 100 μM, respectively, as a positive control. Two hours after sample treatment, LPS were added to the medium with a final concentration of 1 µg/mL and were incubated for 22 h. Then, after 10 min incubation of the supernatant with Griess reagent, OD 540 nm was measured.

### 2.7. Genomic Analysis and Secondary Metabolites Prediction

Genome sequencing was conducted using the MGI platform, with sequencing reads assembled by CLC Assembly Cell 5.1.1. Gene annotation was performed using PATRIC 3.5.36 (https://www.patricbrc.org) [38]. The 16S rRNA gene and genome sequences of strain LN180102 were deposited in GenBank (accession numbers are PV554952 and JBNIIX000000000, respectively). Other LAB strains’ genome sequences were retrieved from NCBI, and phylogenomic trees were constructed on the TYGS online server (https://tygs.dsmz.de), with modifications made via iTOL [39]. Functional metabolite BGCs were analyzed with the bacterial version of the antiSMASH v7.0 [40].

### 2.8. β-Glucosidase and Molecular Docking

The LN180102 genome was analyzed to identify potential β-glucosidase homologs, which could bioconvert geniposide to genipin. BLASTp (https://blast.ncbi.nlm.nih.gov/Blast.cgi, accessed on 18 March 2025) was used for gene sequence alignment under default settings, and the results were ranked by BLAST score, query coverage, and percent identity to identify LN180102_00330 as the most likely homolog. The amino acid sequence LN180102_00330 was submitted to AlphaFold 3, and a predicted 3D structural model was generated [41], which was then converted to PDB format using OpenBabel (v3.1.1). The structure of geniposide (SDF file of the 3D conformation) was obtained from PubChem (https://pubchem.ncbi.nlm.nih.gov, accessed on 18 March 2025). Molecular docking of LN180102_00330 with geniposide was performed using CB-Dock2 (BioLip2 database v2025.04.23), a structure-based molecular docking tool that predicts binding modes and affinities of protein-ligand complexes [42]. This study used CB-Dock2 for docking and scored using the AutoDock Vina (v1.5.7) scoring function (kcal/mol). For the selected binding sites, the results showed a grid center of (−12, −13, 11) Å, a box size of (22, 22, 22) Å, and a cavity volume of 212 Å^3^. The CB-Dock2 web interface only returns the top-ranked pose for each cavity, without providing num_modes or additional conformations. Docking results were visualized and refined using PyMOL (v2.6.0), an interactive tool for manipulating molecular structures to understand protein features and functions, including the binding mode of geniposide and key interactions between the protein and ligand. These methods enabled the identification of potential β-glucosidase homologs in the LN180102 genome and provided preliminary structural and functional insights, laying the foundation for further research into their applications.

### 2.9. Statistical Analysis

Results are expressed as the mean ± standard deviation (SD) of the data presented, and each experiment was independently replicated three times. The data were subsequently analyzed using IBM SPSS 27.0 and plotted using Excel.

## 3. Results

### 3.1. LABs Library Construction and Herbal Medicinal Plants Selection

In total, 191 LABs were isolated and categorized into one order (Lactobacillales), four families (*Lactobacillaceae*, *Leuconostoceae*, *Enterococcaceae*, *Streptococcaceae*), 6 genera, and 36 species. The composition of LABs library on genus level was as follow: *Lacticaseibacillus* (8 strains, 4%), *Lactiplantibacillus* (52 strains, 27%), *Latilactobacillus* (14 strains, 7%), *Leuconostoc* (42 strains, 22%), *Levilactobacillus* (26 strains, 14%), *Limosilactobacillus* (9 strains, 5%), *Enterococcus* (19 strains, 10%), *Mammaliicoccus* (1 strain, 1%), *Pediococcus* (8 strains, 4%), *Secundilactobacillus* (2 strains, 1%), *Streptococcus* (2 strains, 1%) and *Weissella* (8 strains, 4%) based on 16S rRNA sequences blast results. In total, 22 herbal medicinal plants were selected based on the information from ‘*The Dongui Bogam*’, the Medicinal Herbal Materials Bank, patents, and thesis data, all of which contained bioconvertible components with high content (Appendix A). *Gardenia jasminoids* was recorded 4,054 times from 1850 to 2025 in the documents (https://www.gbif.org, accessed on 18 March 2025) (Appendix A). It was introduced to South Korea from China around A.D. 1500, and it is now planted as a garden tree due to its beautiful white flowers, medical applications, and use as a dye in food and textiles (Appendix A).

### 3.2. Bioconversion of G. jasminoides Component

Among the isolates, 14 strains showed the ability to bioconvert *G. jasminoides*, with their closest homologs listed based on 16S rRNA sequences (Table 1). Two similar strains *Levilactobacillus* sp. LN180082 and LN180102 exhibited the highest bioconversion ratio, around 40%, as indicated by changes in HPLC peak profiles at a wavelength of 240 nm. Fermentation of *G. jasminoides* water extracts with *Levilactobacillus* brevis sp. LN180082 and LN180102 resulted in the disappearance of a peak at 13.9 min and the appearance of a peak at 15.2 min. From the HPLC profiles of *G. jasminoides* fermentation with *Levilactobacillus* brevis sp. LN180102, a total of five compounds were isolated, corresponding to peaks P1–P5 (Figure 1A–C). Peak P2 and P3 from *G. jasminoides* extracts showed the same retention times as the standard components geniposide and genipin (Figure 1D,E). Their molecular weights were determined to be as follows: P1 (549.5), P2 (388.5), P3 (226.5), P4 (696.7), and P5 (174.4), as evidenced by the highest intensity peaks detected at specific retention times and ion peaks. P2 and P3 molecular weights were also consistent with the standards geniposide and genipin (Appendix A). Therefore, the bioconversion process involves a glucose unit loss from geniposide and is turned into genipin, and after fermentation, 40% was bioconverted (Figure 1F and Appendix A).

### 3.3. Anti-Inflammatory Enhancement Through G. Jasminoides Bioconversion

The anti-inflammatory activities of *G. jasminoides*, geniposide, and genipin were measured by NO content in LPS-induced RAW 264.7 cells. Both extracts of fermented and non-fermented *G. jasminoides* showed no cytotoxicity up to a concentration of 100 µg/mL. Similarly, geniposide and genipin also did not exhibit cytotoxic effects at concentrations up to 40 uM (Figure 2A). Further assays were conducted using ethanol extracts at 100 µg/mL or single compound concentration at 40 uM to evaluate their effects on NO production. *Levilactobacillus* sp. LN180082 and LN180102 fermentation sample treatment at a concentration of 50 µg/mL, NO content was reduced by 15.2% and 6.9%, respectively. When the treatment concentration was increased to 100 µg/mL, the NO-reducing effect reached up to 30.4% and 24.9%, respectively. Geniposide exhibited no significant effect on NO levels, whereas genipin decreased NO content significantly by 26.2% and 51.8% at treatment concentrations of 20 µM and 40 µM, respectively. (Figure 2B).

### 3.4. Phylogenetic and Morphological Analysis of Levilactobacillus sp. LN180102

Phylogenetic analysis based on 16S rRNA genes (1437 bp for LN180082 and 1420 bp for LN180102) revealed that they were closely related to *Levilactobacillus brevis* ATCC 14869^T^ (Figure 3A), with 99.72% and 99.93% sequence similarity, respectively (Table 1). *Levilactobacillus* sp. LN180102 formed white-colored colonies on MRS agar media and was rod-shaped with a cell size of 0.8 × 3.5 um (Figure 3B,C).

### 3.5. Genome Sequences and Beta-Glucosidase Analysis

The genome of *Levilactobacillus* sp. LN180102 consists of a circular chromosome of 2,473,228 bp, with an average G+C content of 45.3% (Figure 4A). Among the 5441 predicted genes, 2926 were identified as open reading frames (ORFs), of which 2444 were confirmed as protein-coding sequences (CDS). The genome contained 65 predicted RNA genes, including 59 tRNAs, 5 rRNAs, and 1 tmRNA. A total of 65 pseudogenes were identified, and 41 CRISPR loci were annotated by CRISPR/Cas Finder. Molecular docking studies were conducted to investigate the interaction between the protein LN180102_00330 and geniposide. CB-Dock2 docking results revealed a strong binding affinity between LN180102_00330 and geniposide, with a Vina score of −5.6. The binding site, defined by center coordinates (−12, −13, 11), had a cavity volume of 212 and a docking size of (22, 22, 22), ensuring proper substrate accommodation in the active site. These results suggest that LN180102_00330 binds favorably to geniposide, likely enhancing its catalytic efficiency in the bioconversion reaction. Figure 4B illustrates the binding mode of geniposide within the active site of LN180102_00330. As shown, the overall binding mode (left panel) shows geniposide (colored blue and red) positioned within the active site. The magnified view (Figure 4B, right) reveals that arginine-164 (ARG-164) forms hydrogen bonds with geniposide via its guanidine group at 3.1 Å and 3.4 Å, within the typical hydrogen bond range. This observation is consistent with the bioconversion of geniposide to genipin catalyzed by β-glucosidase (Figure 1F). Other residues, such as glutamic acid-416 (GLU-416), interact with geniposide at distances of 1.9 Å, 2.6 Å, and 2.9 Å, further enhancing substrate binding stability. These interactions are crucial for understanding protein function and designing potential drug candidates. The β-glucosidase activity of LN180102 was also confirmed by the esculin assay (Figure 4C).

### 3.6. Probiotic Characteristics

The selection criteria for potential probiotics have been extensively studied and refined over time. Probiotic strains must be capable of surviving harsh conditions such as various temperatures, high salt concentration, acidic environment, and exposure to bile in the human body, so that effectively colonize in the intestinal system then potentially provide benefits to the host. In terms of physiological characteristics, *Levilactobacillus* sp. LN180102 showed broad growth temperature from 10 to 40 °C (optimum 25–37 °C) and high NaCl tolerance up to 7% (optimum 1%). Hemolytic activity was not detected. It just showed resistance to vancomycin, ciprofloxacin, and exhibited no resistance to all other tested antibiotics. *Levilactobacillus* sp. LN180102 showed very weak antimicrobial activity against Gram-positive pathogens, including *Staphylococcus aureus* KCTC 1621^T^ and *Listeria monocytogenes* KCTC 3569^T^, and no activity against Gram-negative pathogens, *Escherichia coli* KCTC 1682^T^, *Pseudomonas aeruginosa* KCTC 2004^T^, and *Cronobacter sakzakii* KCTC 2949^T^ (Table 2). From the genome analysis on antiSMASH, LN180102 was the only one that had a lanthipeptide biosynthesis gene cluster, which may be the reason for antimicrobial activity (Appendix A). Regarding acid tolerance, *Levilactobacillus* sp. LN180102 survived at rates of 79.7% and 75.6% ratio after 2 and 3 h of exposure to pH 2.0, respectively. For bile tolerance, the strain exhibited survival rates ranging from 63.2% to 83.9% after 0.3% bile treatment for 2 to 8 h. Furthermore, treatment with 0.1–0.4% phenol for 24 h did not significantly affect cell growth (Figure 5A–C).

## 4. Discussion

In this study, we systematically carried out research on LABs library construction, candidate screening, microbe identification, genomic analysis, and functional study. Following the GRAS level probiotic standard, we also evaluated physiological characteristics, antibiotic resistance, and tolerance to acid, bile, and phenol of the microbe to ensure their suitability as probiotics for industrial use.

Among the 191 isolates, LN180102 exhibited the strongest bioconversion activity, capable of converting 40% geniposide to genipin. This bioconversion activity was almost twice as high as that of other probiotics, including *Levilactobacillus casei* KFRI 127, *Levilactobacillus curvatus* KFRI 166, *Levilactobacillus confuses* KFRI 227, which only converted 16% [26] (Figure 1A). While the bioconversion activity of LN180102 was lower than that of other enzymes, it offers significant advantages, such as not requiring controlled pH or temperature conditions (Appendix A) [26,27,29,30,31,32,43,44,45,46,47,48].

To our knowledge, this is the first report on the bioconversion of geniposide to genipin through fermentation using *Levilactobacillus* sp. LN180102, which shows a 99.93% similarity in 16S rRNA with *Levilactobacillus brevis* ATCC 14869^T^. Additionally, an ANI (Average Nucleotide Identity) value of 99% was observed, confirming the close genetic relationship between these two strains. This result was corroborated by both 16S rRNA-based phylogenetic analysis and genome-based phylogenomic analysis. However, significant differences were noted in the comparison of β-glucosidases. Two β-glucosidases, LN180102_00330 and LN180102_02111, were identified from the genome annotation of LN180102. BLASTP analysis revealed that LN180102_00330 shared a 98.92% sequence similarity with the reference protein 001755 from *Levilactobacillus brevis* ATCC 14869^T^. Molecular docking analysis further indicated that LN180102_00330 could catalyze the conversion of geniposide to genipin, similar to protein 001755. Both enzymes were shown to interact with the ether bond in gardenia glycoside to initiate the catalytic reaction. However, despite the high sequence similarity, the key catalytic residue, ARG-164, in LN180102_00330 was positioned much closer to the ether bond (approximately 3.1 Å) (Figure 4B) than the corresponding residue in protein 001755 (7.08 Å). This shorter distance likely promotes hydrogen bond formation, potentially enhancing catalytic efficiency during hydrolysis [49].

In contrast, a study on β-glucosidase 1OIF from Thermotoga maritima demonstrated its ability to hydrolyze geniposide to genipin, but the molecular docking model did not identify any key residues interacting with the ether bond [50]. While molecular docking simulations may have inherent errors, the close interaction of ARG-164 in LN180102_00330 with the ether bond suggests that LN180102_00330 may exhibit superior catalytic activity and hydrolysis efficiency. Though this was supported by substrate activity testing using esculin (Figure 4C), indicating LN180102 as a promising candidate for biotechnological applications, such as enhanced yield and rapid conversion.

In addition to bioconversion activity, LAB strains need to meet current food safety regulations if they are not yet classified as GRAS-level strains. As such, individual lactobacilli should undergo antibiotic resistance analysis, β-hemolysis testing, and virulence gene analysis before being considered for industrial use [51]. Probiotic strains must also meet several other criteria to benefit human health. Specifically, they must exhibit high tolerance to salt, temperature, acid, bile, and phenol to survive harsh conditions in the human gut and adapt to the intestinal system [52].

*Levilactobacillus* sp. LN180102 was negative for hemolysis tests. It is resistant to vancomycin and ciprofloxacin among the tested 10 different antibiotics, which were from the standard of the European Food Safety Authority’s (EFSA) Qualified Presumption of Safety (QPS) list [53]. LN180102 showed a broad range of growth temperature (10–40 °C) and high salt tolerance, which was up to 7%, which may be due to the isolation source, kimchi, which is usually made very salty. This strain could be used as a probiotic based on the safety data that we tested above.

Strain LN180102 also showed antimicrobial activity against two Gram-positive pathogens *Staphylococcus aureus* KCTC 1621^T^ and *Listeria monocytogenes* KCTC 3569^T^, possibly due to the production of organic acids or lanthipeptides, even though the gene cluster similarity is very low from the antiSMASH data (Appendix A).

Regarding inflammation, patients often experience side effects with current treatments, prompting the development of new therapeutic agents with fewer toxic effects. Research is underway to identify therapeutic agents with reduced toxicity and side effects, particularly those derived from natural materials such as herbal medicines, which have long been used in folk remedies [54]. Fermentation of G. jasminoides with strain LN180082 resulted in a 5.5–8.2% greater reduction in NO production compared to LN180102. However, considering the antimicrobial activity against *Staphylococcus aureus* KCTC 1621^T^ and *Listeria monocytogenes* KCTC 3569^T^, LN180102 represents a more suitable strain for practical application. Genipin single treatment at a concentration of 40 uM could decrease 9.4% more NO content compared with L-NAME. At a concentration of 40 µM, both geniposide and genipin exhibited slight cytotoxicity, whereas the fermented samples were non-cytotoxic even at concentrations up to 100 µg/mL. So, probiotic-mediated whole-cell bioconversion of geniposide to genipin presents a promising, low-cost, eco-friendly, and safe alternative to enzymatic and chemical methods for enhancing the value of *G. jasminoides*.

## 5. Conclusions

The whole cell bioconversion of geniposide to genipin is demonstrated by *Levilactobacillus* sp. LN180102 through fermentation with *G. jasminoides* water extract, without the need for any additional interventions. This bioconversion process improves anti-inflammatory effects by decreasing NO production in RAW 264.7 cells. The strain LN180102, which is of kimchi origin, demonstrates a wide spectrum of tolerance to a variety of growth conditions, such as high salt, acid, bile, phenol, and temperature fluctuations. Furthermore, it is susceptible to a variety of antibiotics and lacks hemolytic activity. The strain also exhibits antimicrobial activity against *Listeria monocytogenes* KCTC3569^T^ and *Staphylococcus aureus* KCTC1621^T^. These results indicate that *Levilactobacillus* sp. LN180102 has the potential to serve as a probiotic, particularly in the bioconversion of *G. jasminoides*. Consequently, it is a prospective candidate for functional food applications. Additional research is required to investigate the underlying mechanisms and assess the probiotic candidate in the context of other herbal remedies.

## Figures and Tables

**Figure 1 foods-14-04156-f001:**
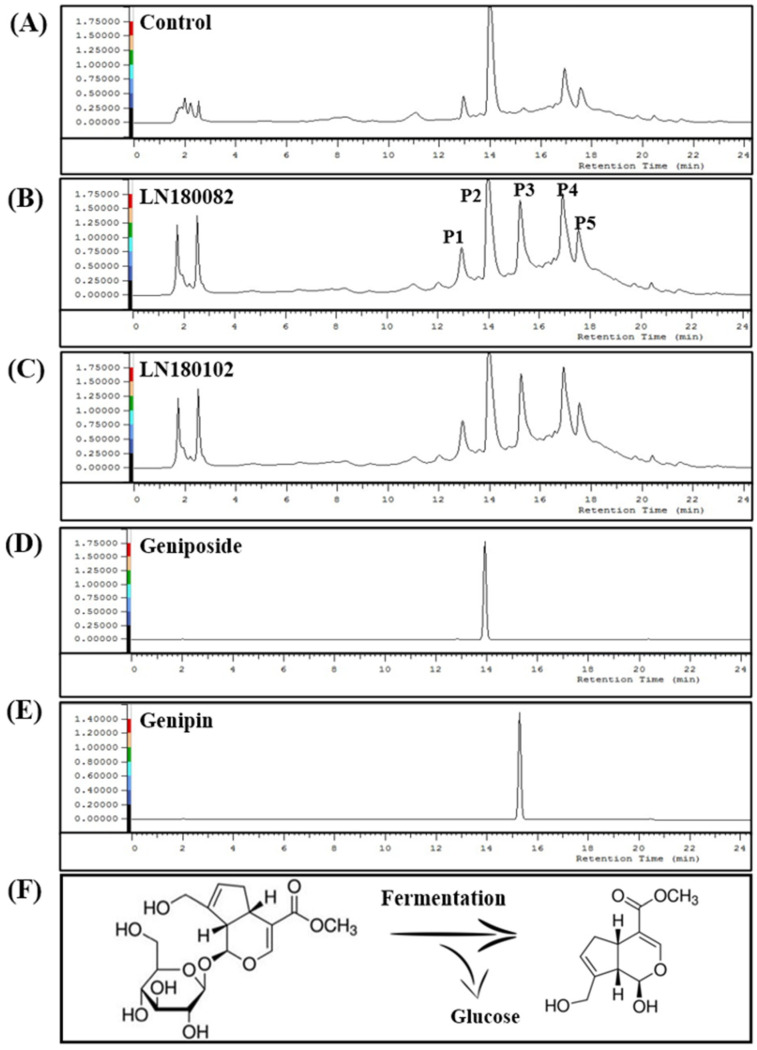
HPLC profile of *G. jasminoides* and its fermentation. (**A**) Control (*G. jasminoides* extract); (**B**) fermentation with LN180082; (**C**) fermentation with LN180102; (**D**) geniposide; (**E**) genipin; and (**F**) bio-conversion scheme.

**Figure 2 foods-14-04156-f002:**
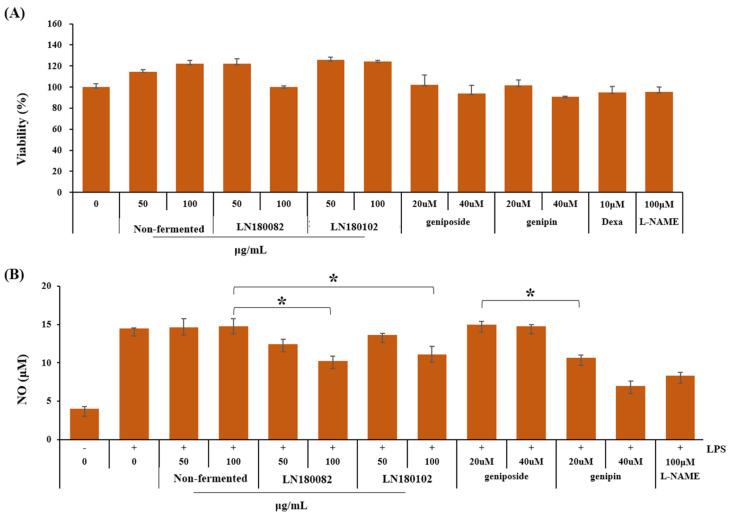
Anti-inflammatory improvement via fermentation of *G. jasminoides*. (**A**) Each component’s cell toxicity, (**B**) NO content. All experiments were independently repeated three times. Single asterisks (*) indicate statistically significant differences between the non-fermented sample treatment group and fermented with LN180082 or LN180102 groups, geniposide and genipin treatment groups at *p* < 0.01.

**Figure 3 foods-14-04156-f003:**
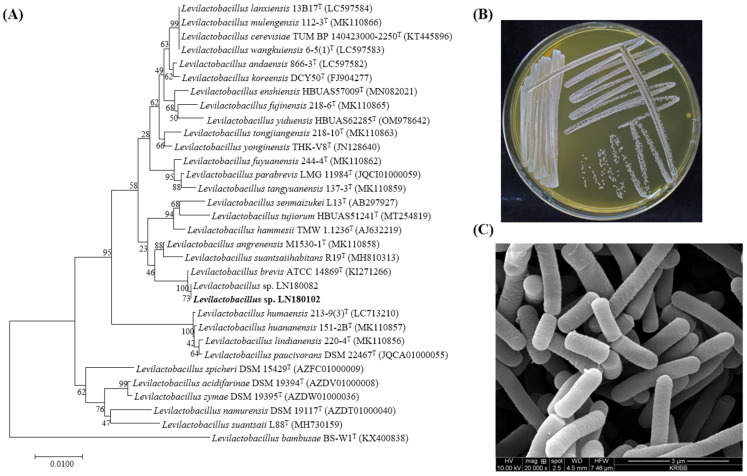
(**A**) Phylogenetic tree of *Levilactobacillus* sp. LN180102; (**B**) cell growth on MRS agar plate at 28 °C for 2 days; (**C**) SEM pictures.

**Figure 4 foods-14-04156-f004:**
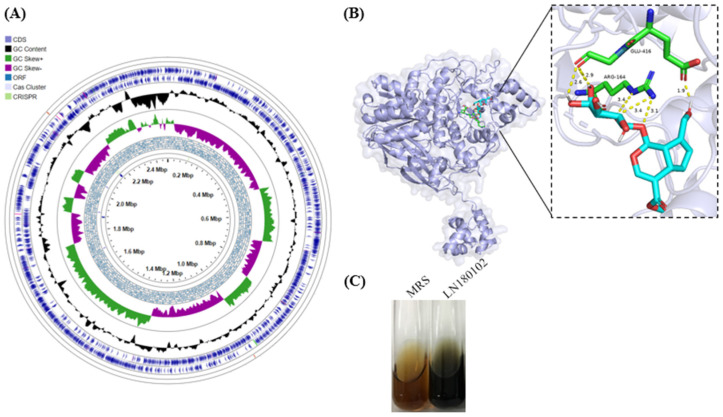
(**A**) Circular genomic map of *Levilactobacillus* sp. LN180102; (**B**) docking analysis of beta-glucosidase with the substrate geniposide; (**C**) beta-glucosidase activity of LN180102 on esculin.

**Figure 5 foods-14-04156-f005:**
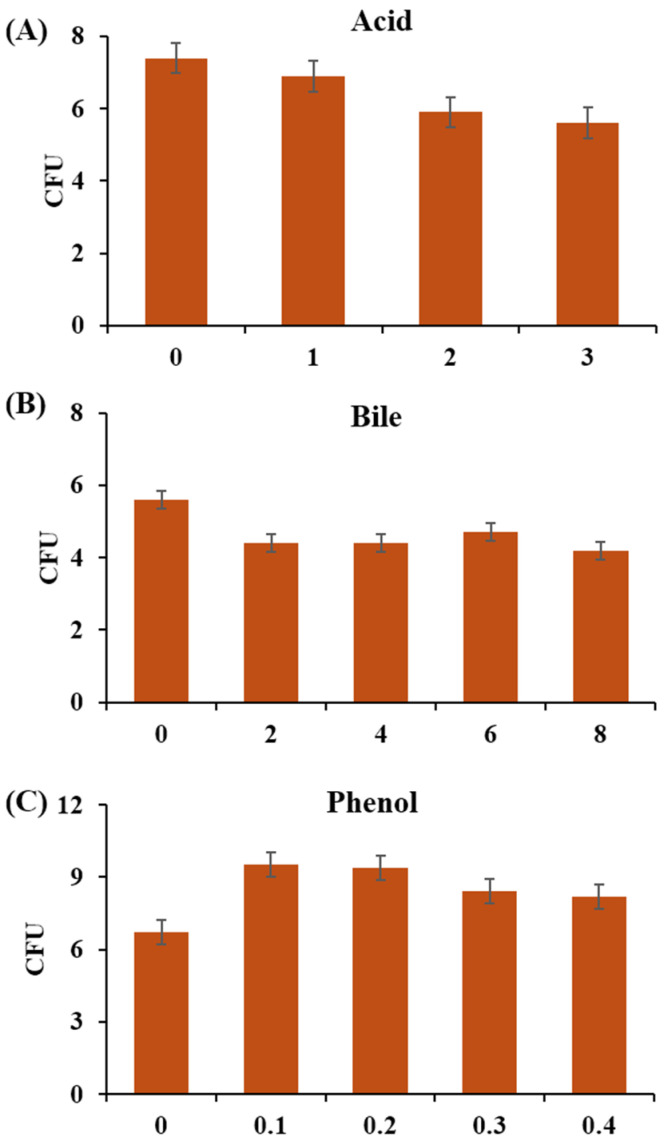
Survivability of *Levilactobacilluis* sp. LN180102 to (**A**) acid; (**B**) bile, and (**C**) phenol treatments. Each test was performed individually in triplicate.

**Table 1 foods-14-04156-t001:** Identification of the bioconversion strains list and their taxonomic name.

No.	Closest Homologous	Similarity (%)	Length (bp)
LN180057	*Lacticaseibacillus rhamnosus* JCM 1136^T^	99.72	1435
LN180061	*Lactiplantibacillus paraplantarum* DSM 10667^T^	99.93	1402
LN180066	*Levilactobacillus brevis* ATCC 14869^T^	99.86	1431
LN180069	*Levilactobacillus brevis* ATCC 14869^T^	99.65	1437
LN180071	*Levilactobacillus brevis* ATCC 14869^T^	99.93	1435
LN180079	*Lacticaseibacillus paracasei subsp. tolerans* JCM 1171^T^	100	1419
**LN180082**	***Levilactobacillus brevis*** **ATCC 14869^T^**	**99.72**	**1437**
LN180087	*Pediococcus inopinatus* DSM 20285^T^	100	1448
LN180088	*Secundilactobacillus malefermentans* KCTC 3548^T^	99.93	1472
LN180090	*Secundilactobacillus malefermentans* KCTC 3548^T^	99.93	1431
LN180092	*Lactiplantibacillus paraplantarum* DSM 10667^T^	100	1419
LN180093	*Enterococcus durans* NBRC 100479^T^	99.79	1462
LN180098	*Leuconostoc mesenteroides subsp. jonggajibkimchii* DRC1506^T^	99.93	1440
**LN180102**	***Levilactobacillus brevis*** **ATCC 14869^T^**	**99.93**	**1420**

**Table 2 foods-14-04156-t002:** Physiology character of LN180102.

General Characteristics	LN180102
Growth temperature (range, optimum) (°C)	10–40, 25–37
NaCl tolerance (%)	0–7
Hemolysis	-
**Antibiotic strip**	**(MIC, μg/mL)**
Ampicillin	1
Vancomycin	R
Clindamycin	0.5
Erythromycin	0.19
Tetracycline	3
Chloramphenicol	3
Ciprofloxacin	R
Gentamicin	3
Kanamycin	12
Streptomycin	24
**Antimicrobial activity**	
*Staphylococcus aureus* KCTC 1621	+
*Listeria monocytogenes* KCTC 3569	+
*Streptococcus mutans* KCTC 3065	-
*Streptococcus salivarius* ATG-P1	-
*Escherichia coli* KCTC 1682	-
*Pseudomonas aeruginosa* KCTC 2004	-
*Cronobacter sakzakii* KCTC 2949	-

## Data Availability

The original contributions presented in this study are included in the article/Appendix A. Further inquiries can be directed to the corresponding authors.

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
