# Peer review of "Bioconversion of Geniposide from Gardenia jasminoides via Levilactobacillus Enhancing Anti-Inflammatory Activity"

_foods, 2025, doi:10.3390/foods14234156_

Round 1

Reviewer 1 Report

Comments and Suggestions for Authors
  1. The introduction provides a solid background, but the novel contribution could be added.
  2. The description of the cell viability and NO assays is incomplete and needs clarification. The source of the RAW 264.7 cells is not mentioned and should be stated. It is also unclear whether the complete medium contained FBS. The authors must specify how the ethanol extracts were dissolved, in what solvent. If ethanol was used, a proper solvent control with the same ethanol concentration must be included to exclude solvent-induced cytotoxicity. The same applies to geniposide, genipin, Dexa, and L-NAME: the solvent for each compound should be clearly indicated, and corresponding vehicle controls described. Finally, the type of control cells used in both the MTT and NO assays (untreated, solvent-treated, or LPS-stimulated only) must be explicitly stated.
  3. The concentration of LPS used for macrophage stimulation is missing and must be specified. Without this information, the experimental setup cannot be interpreted or reproduced. The authors should clearly indicate the exact LPS concentration, treatment duration, and whether all groups (including controls) were exposed under identical conditions.
  4. The molecular docking section lacks essential methodological details. The authors do not specify any CB-Dock2 parameters (grid size, scoring function, or number of poses), nor describe protein or ligand preparation. Without these, the docking results are not reproducible.
  5. The β-glucosidase assay section is incomplete. The methods do not describe how the esculin assay was performed, no substrate concentration, buffer composition, incubation temperature, time, or detection method are given. The authors must provide a full, detailed description of how this assay was conducted to ensure reproducibility and to support the claim of confirmed enzymatic activity.
  6. Figures need clearer captions with experimental details, number of replicates, and statistical significance. Figure 2, for example, lacks explanation of NO data, sample groups, and p-values.
  7. Statistical significance is not mentioned in the text or figure descriptions. If it was not achieved, this should be clearly stated, otherwise, significance levels, tests used, and number of replicates must be provided for all results, and also in the conclusion or everywhere the authors talk about anti-inflammatory results it should be stated if statistical significance was achieved or not.

Reviewer 2 Report

Comments and Suggestions for Authors

The manuscript presents a bioconversion of geniposide to genipin using a newly isolated Levilactobacillus sp. LN180102 from Kimchi. The authors demonstrated bioconversion efficiency (40%), identifying β-glucosidase genes responsible via genomic annotation and molecular docking. Anti-inflammatory activity of the fermented extract was evaluated . The authors found also that strain showed probiotic-relevant characteristics.
The research topic is relevant and timely.
However, there are some issues that need to be addressed. I would suggest the following corrections:
Add supporting citations in the Introduction for genipin’s stronger bioactivity vs. geniposide.

In Results, avoid speculative claims such as "dual functionality enhances potential". You should support such sentences with citation or remove.
In Materials and Methods section, calibration curves or reference standard details are needed.
The entire text needs proofreading. Correct multiple grammatical errors.
All references should follow MDPI citation style.

Author Response

Please see the attatchment.

Reviewer 3 Report

Comments and Suggestions for Authors

The study "Bioconversion of geniposide from Gardenia jasminoides via Levilactobacillus enhancing anti-inflammatory activity" developed an eco-friendly method to enhance genipin production from G. jasminoides using lactic acid bacteria. Among 191 strains from kimchi, Levilactobacillus sp. LN180102 showed the highest conversion efficiency and improved the extract’s anti-inflammatory activity by 28.5%. The strain’s strong probiotic traits highlight its potential for sustainable, health-promoting applications.

The introduction is well written and relevant to the topic. My suggestion is to provide the full name of the plant only at its first mention, as done in lines 37–38 (Gardenia jasminoides Ellis, Rubiaceae), and to use the abbreviated form G. jasminoides thereafter (e.g., lines 56, 61, 81, 85, 91). Please check the remainder of the text and make the necessary corrections accordingly.

The same remark applies to Levilactobacillus. When listing the species (lines 73–76), it is sufficient to write L. casei, L. curvatus, L. confusus, L. antri, and L. plantarum, as it is already clear that these belong to the genus Levilactobacillus.

The materials and methods are described in sufficient detail; however, in my opinion, more information should be provided about the raw material used—the dried fruits of G. jasminoides.

The results are presented concisely, and the discussion is adequate.

Author Response

Please see the attatchment.

Round 2

Reviewer 1 Report

Comments and Suggestions for Authors

I am satisfied with the corrections.